# Research on Design of Emergency Science Popularization Information Visualization for Public Health Events-Taking "COVID-19"as an Example

**Hong Li [1] and Kuohsun Wen [2,*]**

[1] School of Creativity and Design, Guangzhou Huashang College, Guangzhou 511300, China; lihong576@gdhsc.edu.cn

[2] School of Design, Fujian University of Technology, Fuzhou 350118, China

\* Correspondence: khwen@fjut.edu.cn

**Abstract:** This study explores the optimization method of emergency popular science information design elements in public health events, breaks through the traditional design with the designer as the subjective consciousness and proposes an emergency popular science information design method oriented by perceptual narrative. First, relevant research on public health events was carried out to screen out and analyze relevant narrative information elements and image elements, and narrative element divergence tree was established to show evaluation indicators. Second, relevant personnel were invited to evaluate the importance and kansei engineering, factor analysis and other methods were used to establish the correlation evaluation indicators of narrative elements. Finally, the optimization narrative elements of popular science information design were calculated with the fuzzy evaluation method to provide an effective auxiliary role for the visualization design of emergency popular science information. Taking "COVID-19 Event" as an example, the narrative design practice of emergency popular science elements was carried out. According to 313 effective questionnaires, the satisfaction of "COVID-19 event" popular science information elements that adopt the optimization method is relatively high, which verifies the feasibility of this method. The conclusion proves that the perceptual narrative design method can obtain the perceptual identity from the audience and plays a positive role in disseminating emergency popular science information.

**Keywords:** emergency popular science; public health events; information visualization; perceptual narrative design; COVID-19; epidemic spread

## 1. Introduction

On 30 January 2020, the World Health Organization officially listed the novel coronavirus epidemic a public health emergency of international concern (PHEIC) [1]. In the face of the severe epidemic situation, how to effectively respond to public health emergencies is a severe and urgent task. Public health events occur suddenly, and the rapid and wide spread cause great danger and impact to the public, especially for infectious diseases. If there is a lack of scientific information and authoritative sources of interpretation and release, rumors are easily spread, leading to social panic and crisis exacerbation. For example, in the SARS incident in China in 2003, due to inadequate emergency science popularization and the unclear prevention and control of the SARS epidemic, rumors spread everywhere, causing people in some areas to rush to buy salt, vinegar and drugs [2], which brought serious challenges to managing government departments and affected the stability of the social and economic order. The main contribution of this study is visualizing a method of providing popular science information during public health emergencies that could help the public and discovering the differentiated needs of the public, which provides a scientific basis for designing such a method.

In the early stages of public health events, making related science knowledge available through the Internet, TV and new media platforms can alleviate panic, and it has a positive effect on the stability of social order because the public will pay attention to the information and expect effective measures. Emergency popular science information refers to the popularization of science related to public security events, including publicizing safety measures such as emergency plans, protection, risk aversion, self-rescue and disaster reduction. It provides the public with relevant scientific knowledge of events and response methods [3]. It helps the public understand science and view public health emergencies from a scientific perspective. We should constantly improve the scientific literacy of the masses and their ability to respond to emergencies and deal with them to comprehensively improve the anti-risk abilities of society.

Emergency popular science plays a particularly important role in public health emergencies. From the perspective of public health, emergency popular science visualization design that is efficient, easy to understand and in line with public psychological needs can popularize knowledge related to public health events. The traditional visualization of popular science information is dominated by designers' subjective consciousness, ignoring the differentiated needs of the public,, so that different groups have different levels of information,, and the communication effect of popular science information is poor. Successful public health information campaigns can optimize and identify the needs of people in different areas and provide a more accurate and effective scientific basis for popular science information design. It is beneficial to construct a theoretical system for public health emergency prevention and comprehensively improve the ability of the society to respond and deal with public health emergencies and risk resistance abilities.

## 1.1. Definitions of Relevant Concepts

Public health emergencies refer to the sudden occurrence of major infectious diseases, mass unexplained diseases, major food and occupational poisonings and other events that seriously affect public health, possibly causing serious damage to public health [4]. According to the nature, extent of injury and coverage scope, public health emergencies can be divided into four levels: particularly significant, major, large and general. Each level is classified according to the following characteristics: the breadth of transmission, the complexity of hazards, the difference in distribution, the diversity of causes, the comprehensiveness of treatment, etc. [5,6]. For example, public health emergencies that have attracted international attention in recent years include the outbreak of the H1N1 swine flu pandemic, the zka virus epidemic in Brazil, the Ebola epidemic in the Democratic Republic of the Congo and COVID-19.

Emergency science popularization conducted to improve the public's response to emergencies mainly refers to the technology popularization, dissemination and education of relevant sciences. Through broadcasting emergency-related knowledge and methods, the public has the tools needed to respond to the emergency and participate in public crisis events [7,8].

At present, visualization technology can be divided into scientific visualization, data visualization, information visualization and knowledge visualization based on their research object, research purpose, main technology and interactive technology [9]. Visualization design of emergency popular science information is a design method that uses design-related methods to visualize emergency science knowledge. It converts knowledge and information into vivid and interesting graphics that can present relevant information quickly and clearly, thus integrating emergency science popularization and visual aesthetics [10]. The development of data technology provides a new form for the visualization design of emergency popular science information in terms of materials and content. Computer technology is often applied to solve corresponding practical problems through algorithms, but to convert the emergency popular science information into easy-to-understand visual graphics, designers need to extract the information for graphic design, which is different from other art forms. It achieves an easy-to-understand purpose through scientific text,

charts, colors, graphics, visual hierarchies and other expression methods rather than pure artistic expressions.

The real effectiveness of emergency science popularization information design lies in its guiding behavior of persuading the public. In the process of information conversion, designers assume the role of "interpreter" who simplifies popular science knowledge, makes the information presentation more organized and effectively promotes the popularization of emergency knowledge. Therefore, the visualization design of emergency popular science information is a scientific and aesthetically related product.

### 1.2. The Concept of Information Visualization Design

Information visualization includes images, knowledge, science, data and other visualization forms that are presented in a static or dynamic way. Its main significance is to express and to convey difficult abstract information intuitively [11]. As early as the middle of the 19th century, Claude Elwood Shannon, an American mathematician, published an important paper on informatics called "Communication Theory of Secrecy Systems", in which maintained that information could eliminate uncertainty [12]. In the 1970s, Edward Tufte, a professor at Yale University, set up a statistical graphics course and published the first book about information visualization called *The Visual Display of Quantitative Information*, mainly elaborating the viewpoint of transforming statistical charts and quantitative information into information design, which was a pioneering concept of information visualization in the field of design. Isabel Meirelles put forward in the book *Design for Information* that information visualization design should be based on the principle of human-computer interaction. In his book *Information Visualization: Design for Interaction*, British Professor Robert Spence defined visualization as establishing a mental model or mental image of things [13]. In 1989 Jock D. Mackinlay and George G. Robertson first proposed the term "Information Visualization" in English and defined it as the use of computer-supported, interactive visual representation of abstract data to increase users' perception of abstract information [14].

The earliest classic case of information visualization design in the field of public health can be traced back to 1858, when Florence Nightingale, a nurse, wrote a report to the British government, which mainly expressed that the government should carry out medical reform to improve the conditions of battlefield hospitals and save more young lives [15]. She used colorful charts such as rose petals to make data impressive. This form of expression was widely used in the field of information visualization, later known as the "Nightingale Rose Diagram" (as shown in Figure 1).

Another case of visualization design related to public health events is the epidemic map drawn by John Snow. In 1854 cholera broke out in London, England. John Snow carried out research and marked the distribution of cases and the location of well water with visual graphics, intuitively showing the relationship between polluted well water and cholera epidemics [16] (as shown in Figure 2). It effectively sniped at the cholera epidemic and triggered a public health movement throughout Europe, which set a precedent for the research of popular science information visualization design in modern public health emergencies.

Disease and death are the ultimate question that human beings need to face. The image of the leading causes of human deaths are schematically shown in Figure 3. The work presents various causes of death in a tree-shaped bubble diagram where small bubbles in the branches connected by each large bubble more accurately show the main causes of death, such as infectious diseases and "bacteria" [17]. The work adopts three colors: red, orange and yellow, evoking the association with death. Therefore, it can prompt people to re-examine and think about health and hygiene issues.

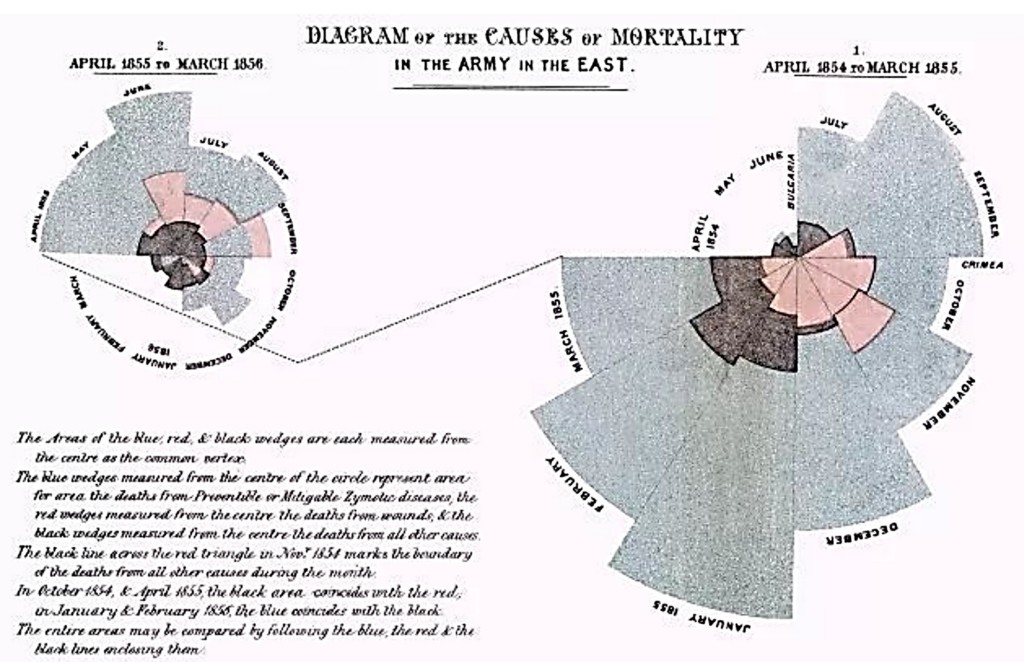

**Figure 1.** Florence Nightingale.

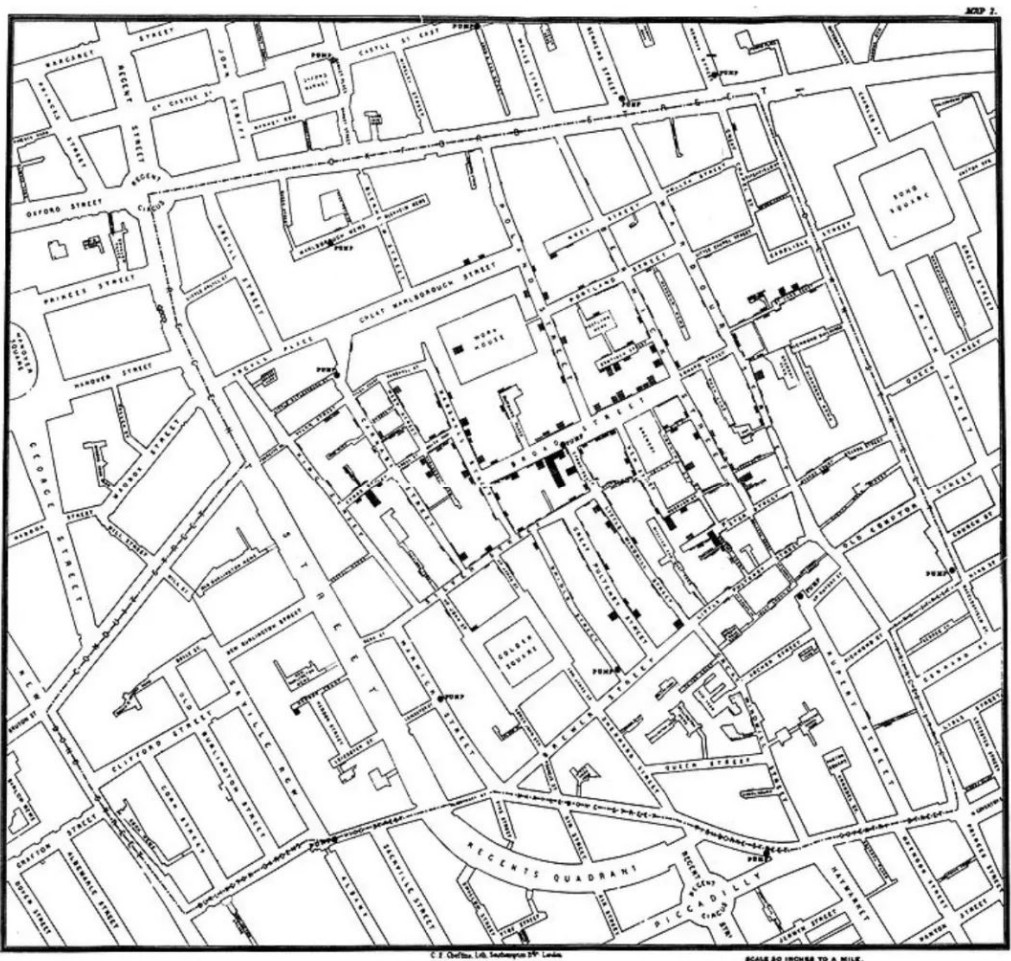

**Figure 2.** Cholera Epidemics (drawn by John Snow).

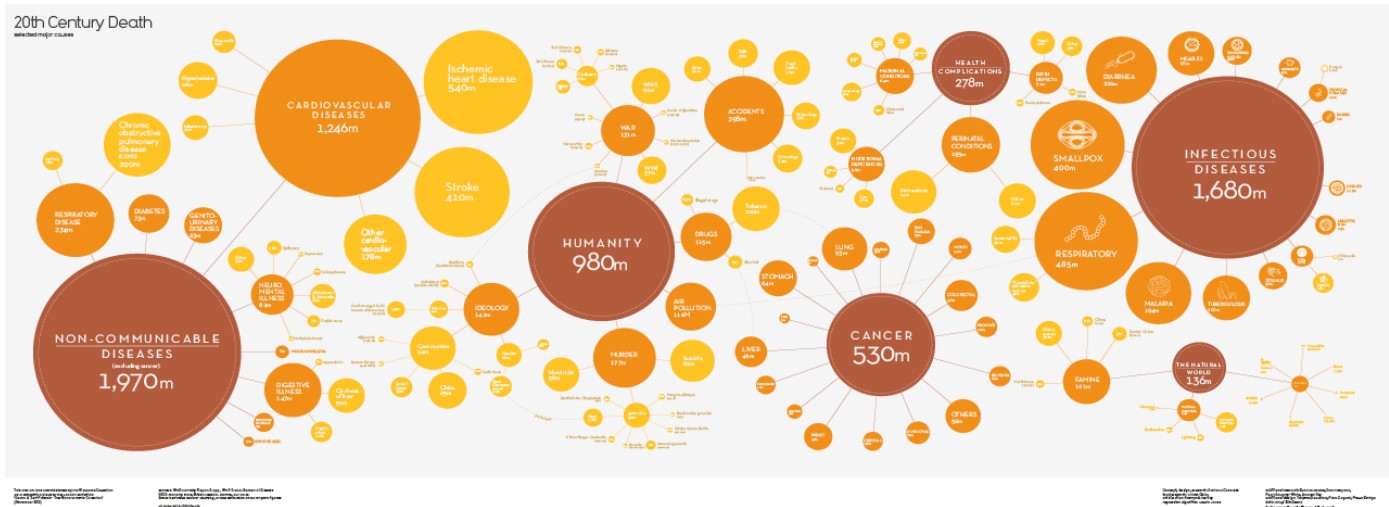

**Figure 3.** Leading causes of human death in the twentieth century.

From a scientific point of view, the above classic public health visualization uses graphical language to convey public health-related knowledge and to disseminate information in a simple and effective visual language, which greatly reduces the audiences' cognitive burden on information. Information visualization design has become one of the most active research fields in biomedicine. It is a kind of method and technology that simplifies abstract concepts, visualizes professional knowledge, visualizes large datasets and displays complex events in a panoramic manner. In public health events, many media use visualization tools to better interpret professional knowledge in the field of public health, which plays an important role in the health propagation process.

## 2. Materials and Methods

### 2.1. Theoretical Basis

The current spread of COVID-19 around the world poses a threat to people's lives and health. Due to the insufficient scientific understanding of this public health incident, many regions and countries have exposed many problems. For example, various pseudo-scientific treatments have emerged in India, using cow urine and cow dung to treat the novel coronavirus. The epidemic in India is out of control due to superstition. It is the lack of scientific knowledge that causes the single-day death toll to be the highest in the world [18]. In terms of COVID-19 prevention and control, China is relatively strict, but a public health science information base with complete and scientific system has not yet been developed, which highlights the weakness of knowledge in emergency science popularization. It also indicates that emergency science popularization mechanisms still need to be strengthened [19]. There are still many problems in the emergency science popularization in response to public health emergencies, which are mainly reflected in the following aspects from the perspective of design and communication. First, there is a lag in emergency popular science knowledge and much information is released after it has affected the public, which means it lacking timeliness. Second, the content and the form of emergency popular science knowledge that is available ignores the perceived demands of the public, is old and is short on aesthetics and attraction. Third, the emergency popular science information is fragmented and not accurate for the target group, resulting in different understanding and acceptance levels of different groups (Figures 4 and 5).

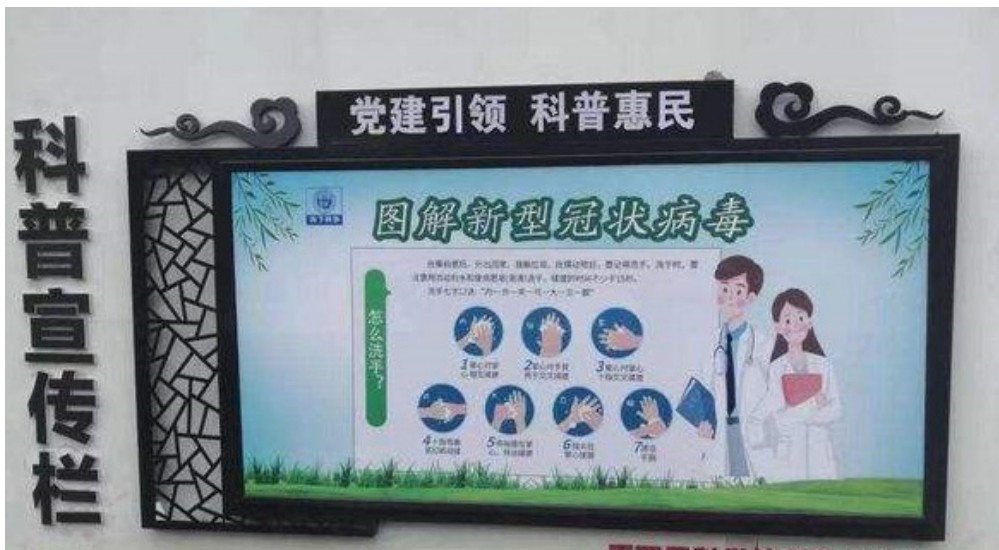

**Figure 4.** COVID-19 science promotion column (Image source network). The picture explains how to wash hands properly to prevent COVID-19.

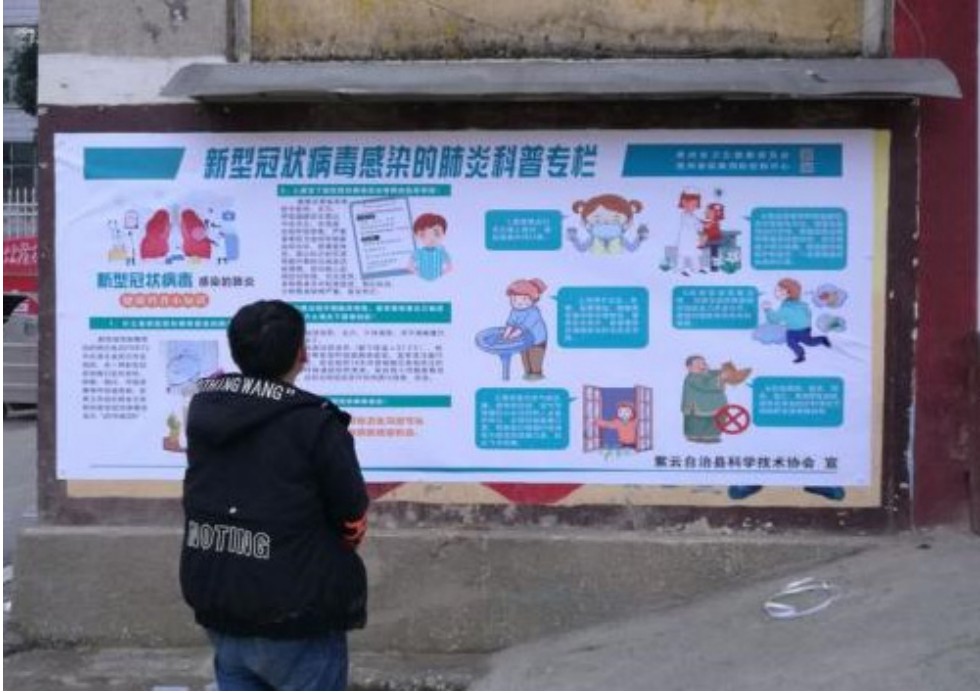

**Figure 5.** A popular science column on COVID-19 infection in the community (Image source network).

From the perspective of perceptual narrative design, this study focuses on the visualization design of emergency popular science information in public health events. Narratology, first born in the French humanities field in the 1960s, used narrative as an information expression way to enable the audience to acquire perception and experience in this cognition [20]. After later development, it was widely used in sociology, geography, journalism, philosophy and other disciplines. In 1973 the American cognitive psychologist L. Standing found that narrative pictures with plots had the best memory effect compared with words and texts. This phenomenon is called the pictorial superiority effect, PSE [21]. Rita Charon, an American scholar, first mentioned the form and the function of narrative medicine in modern medicine in the article "Narrative Medicine: Form, Function and Ethics" in 2001. Narrative visualization, a popular topic for visualization research in recent

years, was developed based on storytelling thought since 2006 [22]. In 2010, Segel and others sorted out the development of narrative visualization in the fields of news narrative, educational media and so on and put forward the design strategy for narrative visualization. In 2010\ WikiLeaks Iraq War Journal: Every Death Map that was published in the British *the Guardian* intuitively and three-dimensionally showed the casualty data related to the Iraq war with narrative visualization, indirectly prompting the withdrawal of British troops from the Iraqi battlefield [23]. In 2012 Lidal and others designed a visualization scheme for geological stories, presenting geological models in the form of storytelling, which helped geological experts draw geological stories quickly. In 2021 Sergey Kashin of Russia created a chart full of tears. This bubble chart animation uses cumulative and confirmed new deaths daily for countries around the world [24]. The application field of narrative visualization has been gradually expanded, including education, literature, journalism, sports competitions and other fields. In particular, it has become one of the most active research fields in biomedicine.

To sum up, scholars at home and abroad have conducted extensive research in the field of narrative visualization design and put forward various narrative design strategies in different fields. However, there are few literature studies on the narrative design of emergency popular science in public health emergencies. According to the current studies, the traditional narrative design is based on the subjective consciousness of designers and ignores the perceptual needs of audiences. As a result, the visual communication of emergency popular science information deviates from an audience psychological model, which causes the audience to lose interest, brings cognitive impairment and affects the dissemination effect of popular science information. To effectively attract the audience's attention and improve their cognition and acceptance of complex information the design of emergency popular science information needs to consider emotional factors, because human decision-making behavior depends not only on rationality but also on sensibility. The persuasion theory model proposed by Aristotle explains the proportion relationship between logical materials and emotional materials (as shown in Figure 6), in which emotional factors occupy an important position of 65% [24]. Logical materials refer to the adoption of specific relevant data, fact analysis and other quantitative evidence, while emotional demands require the participation of the audience and resort to facts and feelings. The optimization narrative design can better resonate through the emotional interaction, so as to attract and persuade the audience and further influence their decisions. Therefore, this study, combining narrative design with Kansei engineering technology, proposes a conceptual design method of popular science information oriented by perceptual narrative to guide designers to conduct emergency popular science information elements optimization design with systematic and scientific methods.

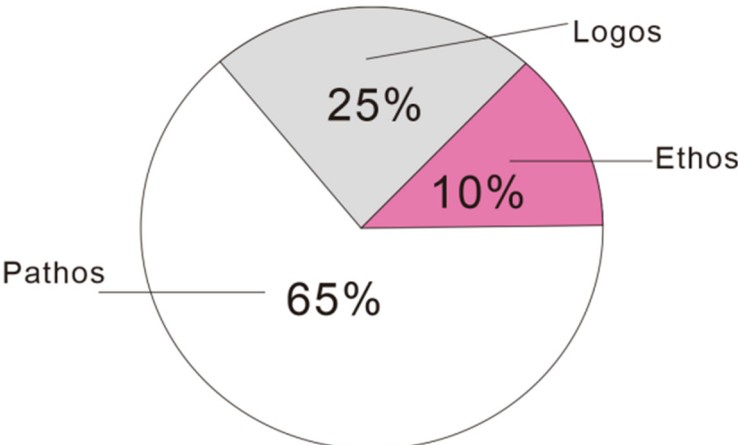

**Figure 6.** Persuasion theory model.

### 2.2. Perceptual Narrative Theory

The essence of narrative design is to reintegrate all the elements of a product with narrative logic. Narrative design of popular science information refers to the process of using design methods to restore, transplant or fabricate emergency popular science information into a related situation and present them in a narrative visualization way. That is, through storytelling the relevant popular science information can be understood, read and remembered easily, and it can be displayed in front of the public more concretely and vividly with the help of graphics. The boring, abstract, rigorous and complicated scientific knowledge in popular science information is presented in the form of graphics, charts and images and is transmitted through visual narration. This kind of narrative design is too subjective for designers to ignore the perceptual needs of users. Perceptual narration is a method that combines narrative design with Kansei engineering technology to evaluate things quantitatively In this approach, the narration can obtain narrative elements that meet the demands of users, and then narrative design can be carried out accordingly [25]. This is an important way to realize the cognitive needs of designers and the perceptual needs of users.

### 2.3. Design Process of Emergency Popular Science Information

Under the background public emergency, the method of perceptual narrative design of popular science information is mainly investigated from the perspectives of epidemiology, public health, psychology, design and other disciplines. To carry out creative and interactive information transmission and to realize the goal of narrating popular science information into a story, it is necessary to balance the relationship between the designer and the audience in the process of narrative design. From the perspective of the whole design process, it can be divided into five basic activities: acquiring information, clarifying the purpose, understanding the audience, perceptual narrative and visual transformation. The basic flowchart of emergency popular science information narrative design is shown in Figure 7.

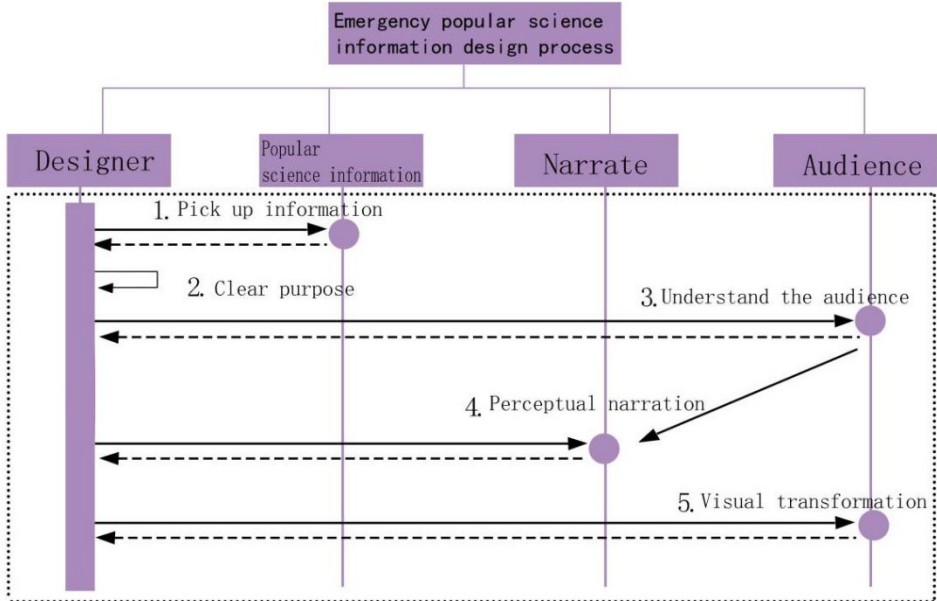

**Figure 7.** The basic flowchart of emergency popular science information narrative design (drawn by author).

**(1) Acquiring information**

First, information visualization narrative design requires information acquisition, which is the process of discovering and refining valuable information. When acquiring information, designers should use statistics or machine learning methods and communicate

with public health experts to get the information about the event, such as the cause, outbreak place, transmission route and so on Then the designer extracts meaningful popular science information [26]. In the early stage of public health emergencies, with all kinds of information disseminated every day and massive information and data distracting people's attention, the public is more eager to know the most urgent problems and countermeasures, which is why designers need to find important information. This is the source of design materials and also provides a basis for public optimization of perceptual elements in the later period.

**(2)    Clarifying purpose**

The visualization design of emergency popular science information aims at popularizing, propagating and guiding the work for the public. Public health emergency is characterized by its suddenness and urgency, so it is necessary to quickly recognize the different visual contents of different stages of the event and accurately target each stage to achieve the best effect [27]. According to the stage purpose of a public health emergency, it can be divided into preventive narrative of public safety, descriptive narrative of prevention and treatment means, prevention policy narrative and standardized treatment narrative. Information visualization narrative is used to explain the causes and solutions of the event to guide the public to establish scientific security awareness, improve emergency prevention and treatment capabilities, and ultimately effectively prevent the spread of the event and the popularization of scientific knowledge.

**(3)    Understanding the audience**

Information narrative design is the process of transmitting valuable information about events. Simple and effective popular science information can help the public make correct judgments and react accordingly. The target audience can be divided into infected patients, suspicious patients, quarantined people, healthy people and other groups. The information visualization narrative should be tailored for different audience groups. For example, people infected with COVID-19 might pay more attention to the information about what drugs can be used to treat COVID-19, while healthy people need to know basic knowledge of the virus, the development of the pandemic and how to prevent being infected and other information. Only by identifying the audience of popular science information and avoiding over generalization, can the purpose of popularization be accurately and effectively achieved.

**(4)    Perceptual narrative**

**Narrative type**
Choosing the appropriate type of data visualization narrative is a crucial step in the whole process. The linear narrative divided by time dimension represents what health events are currently occurring as the timeline changes [28]. The narrative types of public health events can be divided into past tense, present tense and future tense according to the time dimension. The past narrative is used to show the changes of epidemic data in a certain period. The present narrative refers to the current situation and changes of the epidemic, while the future narrative predicts trends of the epidemic.

The three-domain model, first proposed by Peuquet, describes the event semantics with time, location and object, which conforms to the public perception of public health emergencies [29]. The first phase is "What", describing where and what health events occurred—just like telling a story. The second phase is "Why", mainly exploring the dominant potential factors that lead to the current results. In this stage, severity and scientific principles of the epidemic are listed through information visualization narrative, and the truth of the event is conveyed to the public, so that the public can view the event from a scientific perspective and feel assured. The third phase is "Why", mainly exploring the dominant potential factors that lead to the current results. In this stage, severity and scientific principles of the epidemic are listed through information visualization narrative, and the truth of the event is conveyed to the public, so that the public can view the event

from a scientific perspective and feel relieved. For example, the origin of the virus, invasion process, transmission route, symptoms and other information allow the public to know the severity of infection with the virus. The final stage is "How", including the description of the current situation, the explanation of the reason and the solution to the problem, which is communicated to the public through popular science and urges them to act accordingly.

**Narrative elements**

The determination of narrative elements, the focus of the whole design process, can effectively decide the direction of emergency popular science information design. Narrative elements refer to visual symbols with obvious identification characteristics and user perception in the design process. On the other hand, they can guide users to understand visual language through story and thematic features [30]. Taking a popular science product as an example, the narrative elements of the product can be summarized as a set $M$, expressed as $M = Tree(Mc, Mk)$ in the form of a divergent tree graph, among which $Mc$ represents the narrative science information set, $C = 1, 2, \cdots, n$. $Mk$ represents the emotional narrative image set, $K = 1, 2, \cdots, n$. $Mc$ narrative element tree structure diagram [31] is shown in Figure 8. Narrative science popularization presents scientific knowledge about health, while perceptual narrative is to convert information into perceptual images of graphics and express perceptual words with adjectives.

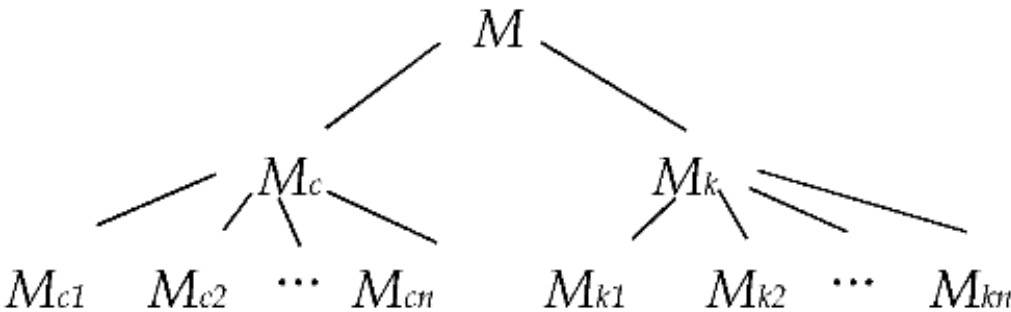

**Figure 8.** The correlation evaluation indicators of narrative element tree structure diagram.

Through the above analysis of perceptual narrative theory, it is found that to explore the elements of popular science information that the audience are interested in, the key lies in establishing the meeting point between popular science information and audience perception. The first step is to analyze the theme form, presentation way and the audience of emergency popular science that can be organized and expressed with a divergent tree graph. The second step is to construct the correlation evaluation index of emergency popular science information elements in the qualitative reasoning way from the perspective of optimization of emergency popular science elements. The third step is to establish a semantic difference scale for experimental samples according to the above perceptual evaluation indexes and use factor analysis to determine the correlation evaluation elements and factor weight coefficient set.

**Optimize the narrative elements of emergency popular science information**

The documents and materials related to emergency popular science information elements are obtained from text mining and natural language. However, the number of information elements obtained from natural language is massive, and the contents about emergency popular science information are boring to the audience. Therefore, it is necessary to evaluate the relevance of emergency information elements and to locate popular science information elements that the audience are interested in. Fuzzy comprehensive evaluation method, based on fuzzy mathematics, is used to optimize factors. The application of fuzzy comprehensive evaluation method is based on fuzzy set theory. The concept of fuzzy set theory was put forward by American automatic control expert L.A. Zeh in 1965 as a method to express the uncertainty of things. The comprehensive evaluation method transforms qualitative evaluation into quantitative evaluation according to the membership degree theory of fuzzy mathematics, that is, using fuzzy mathematics to make an overall evaluation

of things or objects restricted by many factors. It can solve vague and hard-to-quantify problems and is suitable for all kinds of nondeterministic problems.

First, define the perceptual comment set $V$ = (totally consistent, very consistent, generally consistent, not consistent, very inconsistent) and assign $V$ = (5, 4, 3, 2, 1). Then, set up the correlation evaluation factor set $U$ and evaluate all the factors in it to get the upper affiliation matrix $R_1, R_2, \ldots, R_n$ of $U_1, U_2, \ldots, U_n$ and $U_n$'s n-level indicator perceptual comment set $V$. Then set the factor weight coefficient set as $Y$, perform fuzzy linear transformation on $Y$ to obtain fuzzy vector $X$, then $X = Y \cdot R = (a_1, a_2, \ldots, a_n)$. Then, assuming that the fuzzy comprehensive evaluation set of narrative science information elements in item i is $X_i$ and $Vj$ is the perceptual evaluation value corresponding to each evaluation value $Xi$ ($a_1, a_2, \ldots, a_n$), then $M$ is a comprehensive score and $M = \frac{\sum (X_i \times V_j)}{\sum V_j}$. Finally, select the emergency popular science information elements that the audience can identify with based on the comprehensive score of all the narrative popular science information elements $M_1, M_2, \ldots, M_n$. The optimization method of emergency popular science information narrative elements is shown in Figure 9.

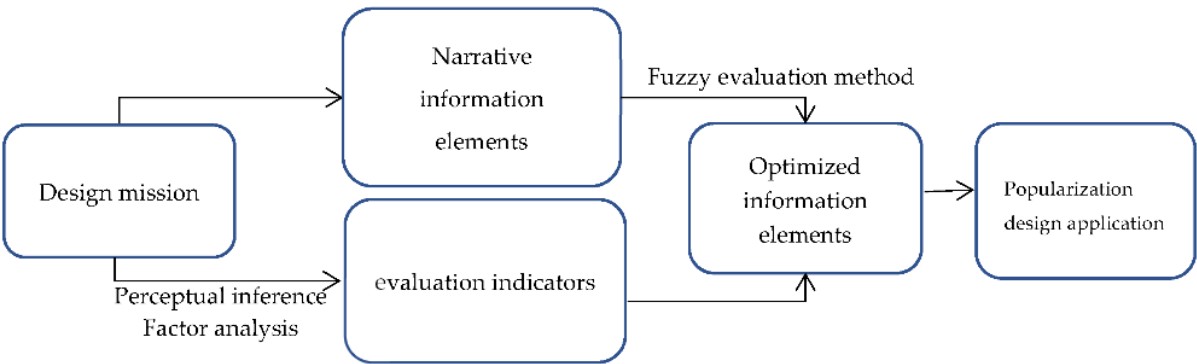

**Figure 9.** The optimization method of emergency popular science information narrative elements.

**(5)     Visual transformation**

Through visual salience research, psychologists found that visual characteristics such as size, direction and color of a figure could attract people's attention preferentially. Designers need to convert the optimized elements of emergency popular science information into visual symbols, a kind of easy-to-understand visual language, so that the audience can master the popular science information conveyed in the visual picture.

There is a lot of popular science information in public health events, and much of it is professional medical knowledge. By using the aesthetic sense of design to eliminate the boredom and fear of medical information theory and with the help of rational graphics and simplified text, the communication purpose of design can be achieved. The basic elements of the emergency popular science design are point, line, surface and color. For the point, we can design its position, size, shape and color attributes, for example, to represent the time, spatial location, arbitrary object and transmission path of the epidemic. Lines include straight lines, broken lines, curves, dotted lines, etc. and different kinds of them can be used to express the mutual relations among various objects such as visual hierarchy, density, reference, guidance, etc. [32]. Lines are helpful to enhance visual direction and order sense in visual narrative design and are helpful for picturing the sense of hierarchy. Surface here refers to the specific image, that is, modeling such as geometric form, cartoon modeling, character modeling and other refined artistic forms such as cartoon modeling of virus images and anthropomorphic modeling, etc. The contents of popular science information or the meaning behind it can be expressed further on the condition that the image is concise and intuitive, and the narration process is in the form of story.

An excellent visualization design work of popular science information usually integrates the knowledge of medicine, statistics, design and so on. Without medical knowledge

as a basis, visualization work will be nothing more than an art illustration. If there is only medical knowledge without the transition of design, it will be a pile of obscure text information. The text, data and other difficult content in popular science knowledge are processed and designed according to visual logic The knowledge information is then transformed into vivid and interesting graphics, and the relevant information is presented quickly and clearly to achieve the high integration of emergency popular science and visual aesthetics. It is an important means to achieve effective communication, to enhance the public's scientific cognition of the event and improve the public's awareness and behavior of preventing public health emergencies.

## 3. Results

### 3.1. Case Design

In accordance with the theoretical study oriented by perceptual narrative mentioned above, the public health event "COVID-19" was selected as the design case in this design practice. It follows the popular science information design process and conducts design evaluation and verification after being completed to maximize the reliability of the conclusions.

At present, COVID-19 is spreading around the world, causing confusion and panic among people in all the countries. How to improve the public's awareness of COVID-19 and how to improve the prevention capacity and comprehensive management level when the epidemic occurs was the goal of the design. This study took COVID-19 as publicized in community as the theme and designed a visualization work of emergency science popularization information, whose target group was S Community, Zengcheng District, Guangzhou, China. There are 3720 families in this community, 27.6% of whom have a junior or senior high school education and 35% of whom have a college education. We used the methods discussed above to carry out conceptual design and popularize the knowledge of COVID-19 to the target group in order to improve the public's scientific understanding of the epidemic.

### 3.2. Analysis of COVID-19 Narrative Elements

For COVID-19, many information elements need to be obtained through multiple channels. This information collection mainly obtained real information from professional websites, academic papers, medical books, expert videos, etc. Based on the COVID-19 information released by the World Health Organization (WHO) and combined with the authoritative information distributed by Chinese Center for Disease Control and Prevention and the Health and Epidemic Prevention Department of the National Health Commission of China, the related professional knowledge was screened to be better understood. As many popular science information elements and image elements related to COVID-19 events as possible were collected and used to build a divergent tree shown in Figure 10.

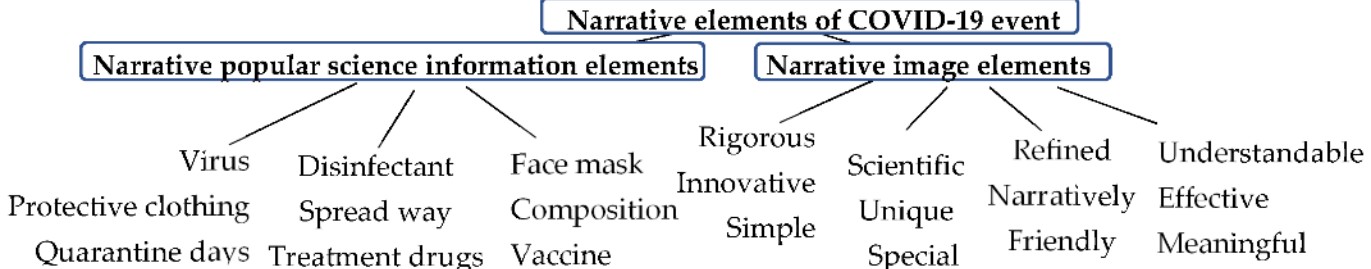

**Figure 10.** Narrative elements of COVID-19 event.

### 3.3. Composition of Evaluation Indicators

The expert evaluation method was used to select people with design experience to make the results more realistic. Ten designers with visual communication design experience

were recruited to form a team that made perceptual inference on the selection of COVID-19 emergency popular science information elements (as shown in Figure 7) [33]. The research group made the next inference with the title of "What design points should be considered in order to optimize popular science information elements that are of interest to the public". From the conclusion, the optimization elements of emergency popular science information visualization were expressed from three aspects: popular science expression, modeling goal and audience perception. Centering on the topic of "what indexes should be considered in order to accurately realize the popularization purpose", we drew a conclusion that scientific, universal, rigorous and other indexes should be considered. We used the same method to infer and get the specific evaluation indicators from the perspectives of modeling goals and audience perception [34]. Finally, we deduced all the evaluation indicators to obtain perceptual judgment elements, as shown in Figure 11.

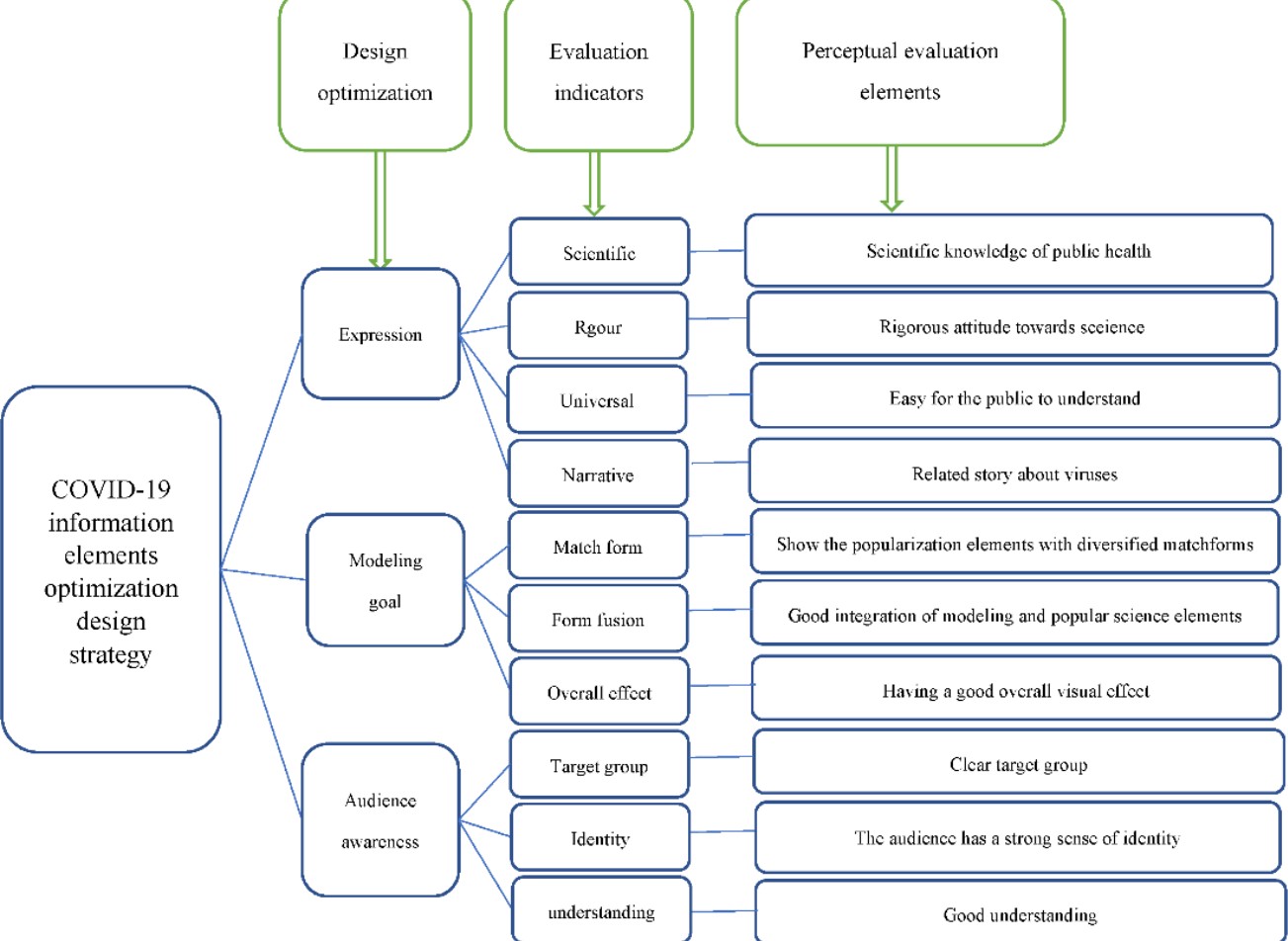

**Figure 11.** COVID-19 perceptual evaluation indicators inference graph of emergency popular science information design.

Based on the evaluation indicators of COVID-19 science popularization elements constructed above analyzed the weight of evaluation elements. First, a questionnaire was set with the theme of "COVID-19 Emergency Information Design Evaluation Indicator Importance Questionnaire". A total of 10 evaluation indexes were designed, and the questionnaire was designed in the form of a Likert scale. Second, 163 questionnaires were distributed to the expert group and the marketing group, and 145 valid questionnaires were collected. With SPSS to conduct KMO value and Bartletts test on the questionnaire results, we found that KMO = 0.673 (>0.6), Sig = 0 (<0.01), and the reliability test result was 0.746, indicating common factors among correlation matrices of COVID-19 popular

science element evaluation indicators suitable for factor analysis [35]. Then, the principal component analysis method was used to extract the common factor, and the scree plot was obtained as shown in Figure 12. Comparing the rotated component matrix with the maximum variance, the eigenvalues of the first four factors visible in the figure were greater than 1, from which four common factors can be extracted.

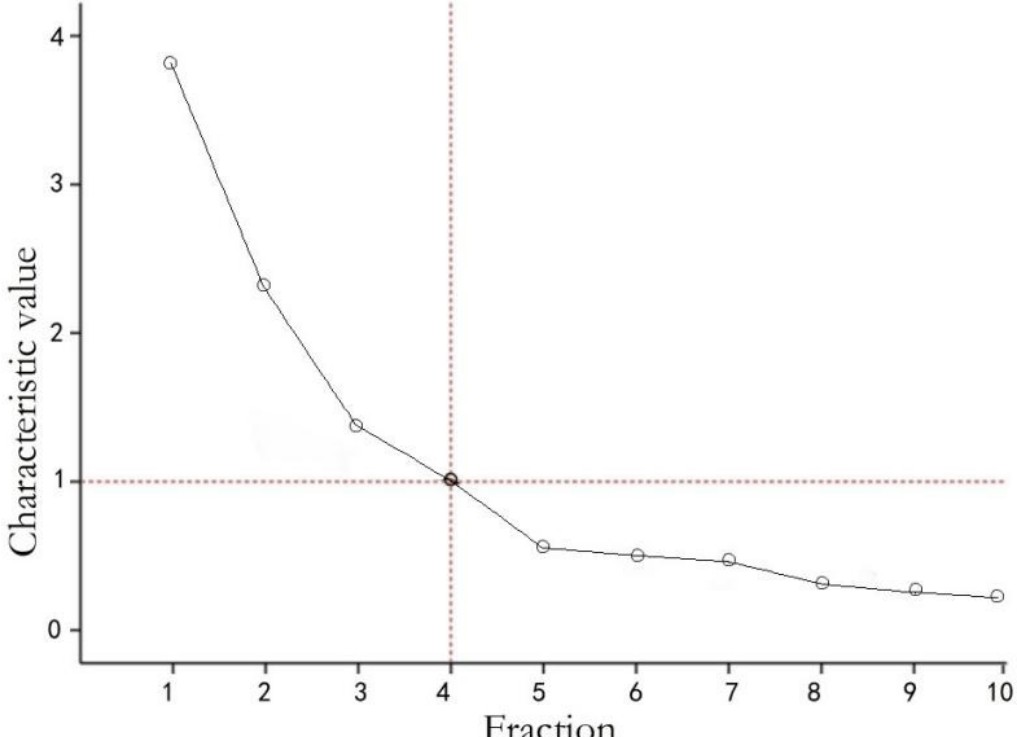

**Figure 12.** Scree plot.

It can be seen from Table 1 that according to the orthogonal rotation factor load matrix, the cumulative variance contribution rate of the first four common factors was 76.953%, and the information loss was less than 23%, indicating that the factor extraction and interpretation ability was good, which met the experimental requirements. Four common factors were positioned respectively. The first common factor $Y_1$ was mainly related to the knowledge, principles and methods of COVID-19 popular science, so it was defined as the significance factor. The second common factor $Y_2$ was mainly related to the design modeling effect in information visualization, so it was defined as the design performance factor. The third common factor $Y_3$ was mainly related to the audience's understanding degree, interest and attraction for popular science information, which is defined as audience expectation. The fourth common factor $Y_4$ was mainly related to the overall visual effect of information visualization and it was defined as artistic appeal factor. The coefficients of four common factors and subordinate factors were calculated to obtain the factor weight coefficient set $W = \{0.516Y_1, 0.179Y_2, 0.154Y_3, 0.139Y_4\}$.

**Table 1.** Factor analysis of 10 evaluation factors.

| Common Factors | Evaluation Factors | Factor Loading | Percentage of Variance | Cumulative Variance Contribution % |
|---|---|---|---|---|
| Significance | Convey scientific knowledge of prevention and control | 0.845 | 38.401 | 38.258 |
| | Popularizing scientific principles | 0.647 | | |
| Design performance | Scientific disposal | 0.832 | | |
| | Characteristic form effect | 0.796 | | |
| | Good modeling transformation | 0.682 | 15.079 | 54.872 |
| Audience expectation | Easy to understand | 0.804 | | |
| | Trigger audiences' interest | 0.859 | | |
| | Design is attractive | 0.738 | 12.326 | 64.239 |
| Artistic appeal | Good visual effects | 0.872 | | |
| | Aesthetic interest | 0.715 | 11.147 | 73.473 |

The COVID-19 science popularization information was initially screened by the research group, and 10 information elements were selected as experimental samples from the preliminary research, which were recorded as $f_1, f_2, f_3, \ldots, f_9$ and $f_{10}$ respectively. We used four defined common factor evaluation indicators, "Significance, Design Performance, Audience Expectation, Art Appeal", to give a score, and assign a value to $V = (5,4,3,2,1)$. The higher the score, the more important the index is [36]. Invited members of the research group scored each element and calculated the membership matrix of each sample according to the score, as shown in the formula below.

$$R_1 = \begin{bmatrix} 0 & 0 & 0.05 & 0.17 & 0.76 \\ 0 & 0 & 0.16 & 0.19 & 0.60 \\ 0 & 0.12 & 0.15 & 0.21 & 0.63 \\ 0 & 0 & 0 & 0.29 & 0.71 \end{bmatrix} \cdots R_{10} = \begin{bmatrix} 0 & 0.08 & 0.26 & 0.31 & 0.49 \\ 0 & 0.08 & 0.06 & 0.22 & 0.62 \\ 0 & 0.17 & 0.16 & 0.39 & 0.42 \\ 0 & 0 & 0.05 & 0.40 & 0.63 \end{bmatrix}$$

Among them, $R_1, R_2, \ldots, R_{10}$ correspond to $f_1, f_2, \ldots, f_{10}$ in an orderly manner.

Based on the above membership matrix f and factor weight coefficient set $W$, the fuzzy vector could be calculated:

$B_1$ = (0, 0.04, 0.19, 0.26, 0.71) $\cdots$

$B_{10}$ = (0, 0.03, 0.25, 0.31, 0.57)

According to the fuzzy comprehensive evaluation set and the perceptual evaluation set, the comprehensive perceptual evaluation value of each narrative element was calculated [37], as shown in Table 2. At the end, the information elements were arranged according to the comprehensive evaluation value and the perceptual evaluation result was $f_2 > f_4 > f_9 > f_6 > f_1 > f_{10} > f_7 > f_5 > f_3 > f_8$. The information element with the top comprehensive evaluation value was the popular science information element that met the demands of users [38]. Therefore, the above sorting information elements are regarded as the optimization elements for COVID-19's popular science information visualization design. In this study, the top four elements including virus, disinfectant, infectious method and mask wearing were selected as the optimization information elements to design the emergency science information map [39]. In addition, visual transformation of narrative design was carried out for popular science information elements, as shown in Table 3.

**Table 2.** Comprehensive evaluation value of COVID-19 popularization elements.

| Information Elements. | $f_1$ Epidemic Prevention | $f_2$ Virus | $f_3$ Drugs | $f_4$ Disinfectant | $f_5$ Taking Temperature | $f_6$ Mask | $f_7$ Protective Clothing | $f_8$ Temperature | $f_9$ Infection | $f_{10}$ Vaccine |
|---|---|---|---|---|---|---|---|---|---|---|
| Comprehensive evaluation value | 4.18 | 4.31 | 4.12 | 4.27 | 4.13 | 4.20 | 4.15 | 4.07 | 4.22 | 4.16 |

**Table 3.** COVID-19 popularization information elements narrative design transformation table.

| COVID-19 Information Elements | Narrative Information | Design Expression |
|---|---|---|
| **Virus** | The structure of COVID-19 virus is in a spherical coronal state, and its internal structure is an RNA virus with envelope on the surface and protrusion on the outside of the envelope, which looks like a crown, so it is called novel coronavirus. |  |
| **Disinfectant** | Novel coronavirus remains relatively weak in the outside world and can be killed with 75% ethanol, chlorine-containing disinfectants and other disinfectants. |  |
| **Infection** | Transmission routes include respiratory droplet transmission and contact transmission. Transmission through respiratory droplets, such as sneezing, coughing, foaming and close contact of exhaled air, can lead to continuous transmission of the disease. |  |
| **Mask** | It's better to choose surgical masks and N95 masks. Others such as sponge masks, activated carbon masks, paper masks, etc., cannot effectively prevent the transmission of novel coronavirus. |  |

The information architecture of the COVID-19 science popularization and prevention infographic was designed from two aspects: narrative path and information content (as shown in Figure 13). First, the principle of narrative design was applied in the infographic architecture. Starting from the development history of the coronavirus, the visual flow can be viewed from top to bottom and the information level is clear, which increases the audience's understanding of information [40–42]. Second, the information includes the development process of coronavirus, virus structure, disinfection methods, infection methods and how to choose masks. Third, the scatter composition was adopted. In the center of the picture, we used circle to represent the novel coronavirus and its structure is shown in the form of close-up so that the audience can read the visualization information comprehensively [43]. Fourth, in the visual conversion stage, blue and purple were used as the overall tone, graphics were designed in a flat way and straight lines, bar charts and geometric illustrations were used to enrich the picture. Fifth, according to the principle of gestalt psychology proximity, the overall image was divided into several block areas to increase the visual hierarchy and to make the small units that are close to each other appear to be a whole [44–46], which enables the audience to search for the interested areas quickly and accurately. It passes on different levels of information to the audience and improves memory. The design sketch is shown in Figure 14.

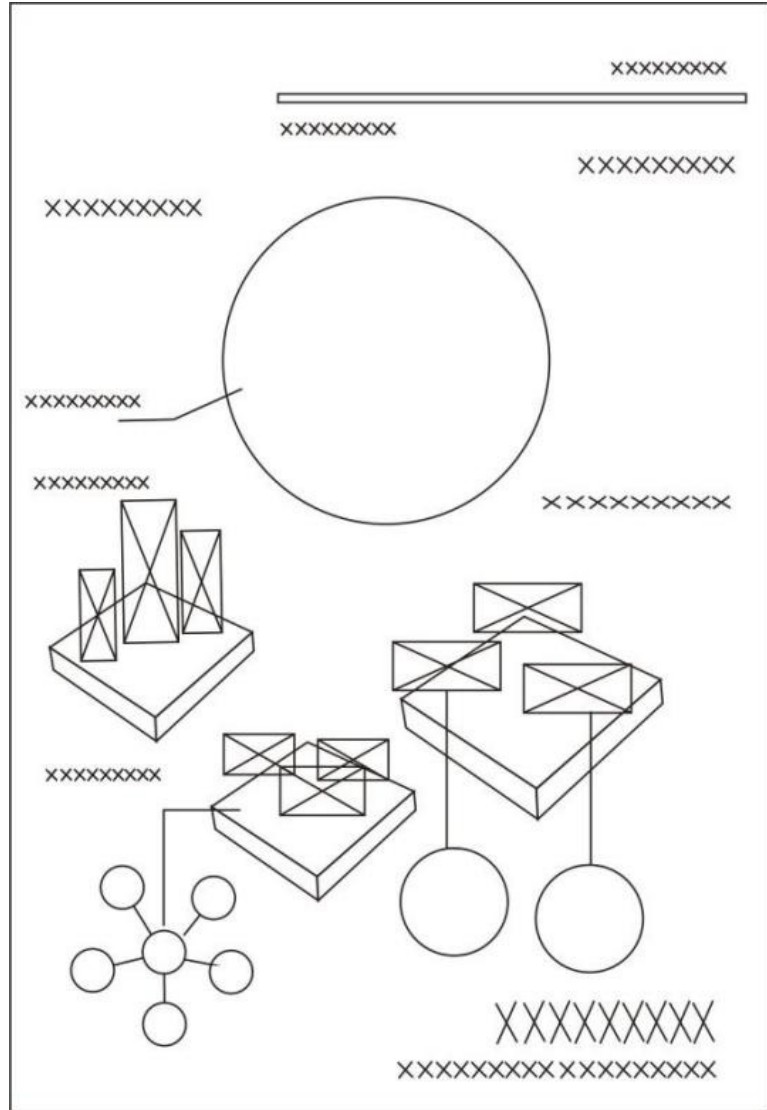

**Figure 13.** COVID-19 Science popularization prevention infographic visual architecture.

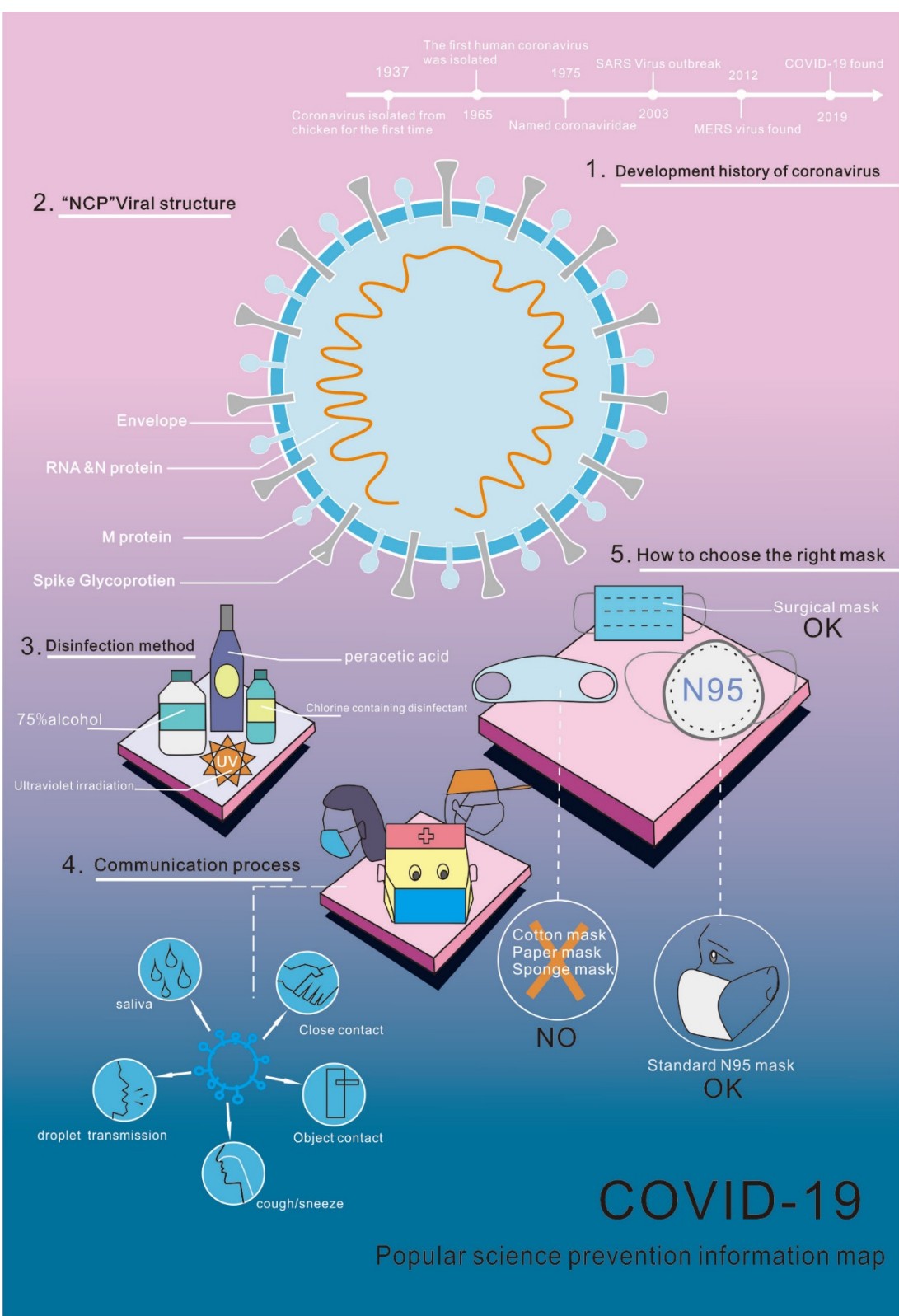

**Figure 14.** COVID-19 science popularization prevention infographic design sketch.

## 4. Discussion

### Design evaluation and conclusions

The purpose of verifying the above popular science information design drawing was to evaluate whether the optimization elements of COVID-19 popular science information met

the perceptual demands of community audiences. The COVID-19 science popularization and prevention infographic designed by the author has been displayed in S Community, Zengcheng, Guangzhou (as shown in Figure 15). A total of 370 people of different ages were randomly invited in the community to score the design and evaluate the design drawing from four aspects: the significance of popular science information, design performance, audience expectation and artistic appeal. A total of 370 questionnaires were distributed and 313 valid questionnaires were recovered. The level 5 satisfaction score of the Rickett scale was recorded as 5, 4, 3, 2 and 1, respectively, corresponding to "very satisfied" "satisfied" "not necessarily" "not satisfied" and "very dissatisfied". According to the statistics of final recycled valid questionnaires, four average satisfaction indicators of the COVID-19 science popularization prevention infographic all exceeded 4.0, as shown in Table 4, which indicates that the design with perceptual narrative method can earn higher satisfaction and the design should also meet the emotional needs of users based on the communication of popular science knowledge [47–49]. Consequently, the outcome meets the requirement of personalized need-based design for different people as well as different regions to achieve the purpose of popularizing science knowledge in the community [50,51]. Compared with the traditional popular science knowledge design, the perceptual narrative design method avoids focusing on the designer's subjective consciousness. Instead, it takes into account the psychological feelings and needs of the audience. This method enables the audience to actively accept popular science publicity [52,53]. Through practice, it has been proven that the perceptual narrative design method is feasible and effective in the popular science application of public health events. It provides a theoretical basis and reference value for the popularization design of public health emergencies in the future [54].

In all kinds of public emergencies, the design of emergency popular science information is an important factor of public health cognition, health attitude and health behavior.

Audiences may have different cognitive feelings in public health emergencies due to the different categories of events, different regions, and cultures. Based on the method of this study, the changes in the above factors should be considered, and the corresponding steps and methods have to be adjusted and modified to make the emergency popular science information design more suitable for local needs. For this study, there are still some deficiencies in terms of experiments such as the insufficient number of selected experimental popularization elements, and the deficiency of group search in perceptual research. This study aims to solve the problem of subjectivity and inaccuracy of the design of traditional emergency popular science information and to achieve the purpose of effectiveness and operability of emergency popular science information design. In the end, based on the popular science practice path including information design dimensions, design process, design methods, design elements, and design evaluation aspects of research, this paper explores the feasibility of its application and design practice, the universal law of science information design approach and the creation of the public health emergency science information visualization-oriented design patterns. Then, the standardized principle of popular science information visualization design is established.

**Table 4.** Evaluation value of COVID-19 science popularization information design drawing.

| Indicators | Significance | Design Performance | Audience Expectation | Artistic Appeal |
|---|---|---|---|---|
| **Satisfaction COVID-19 prevention info-graphic** | Average value | Average value | Average value | Average value |
| | 4.26 | 4.31 | 4.28 | 4.33 |

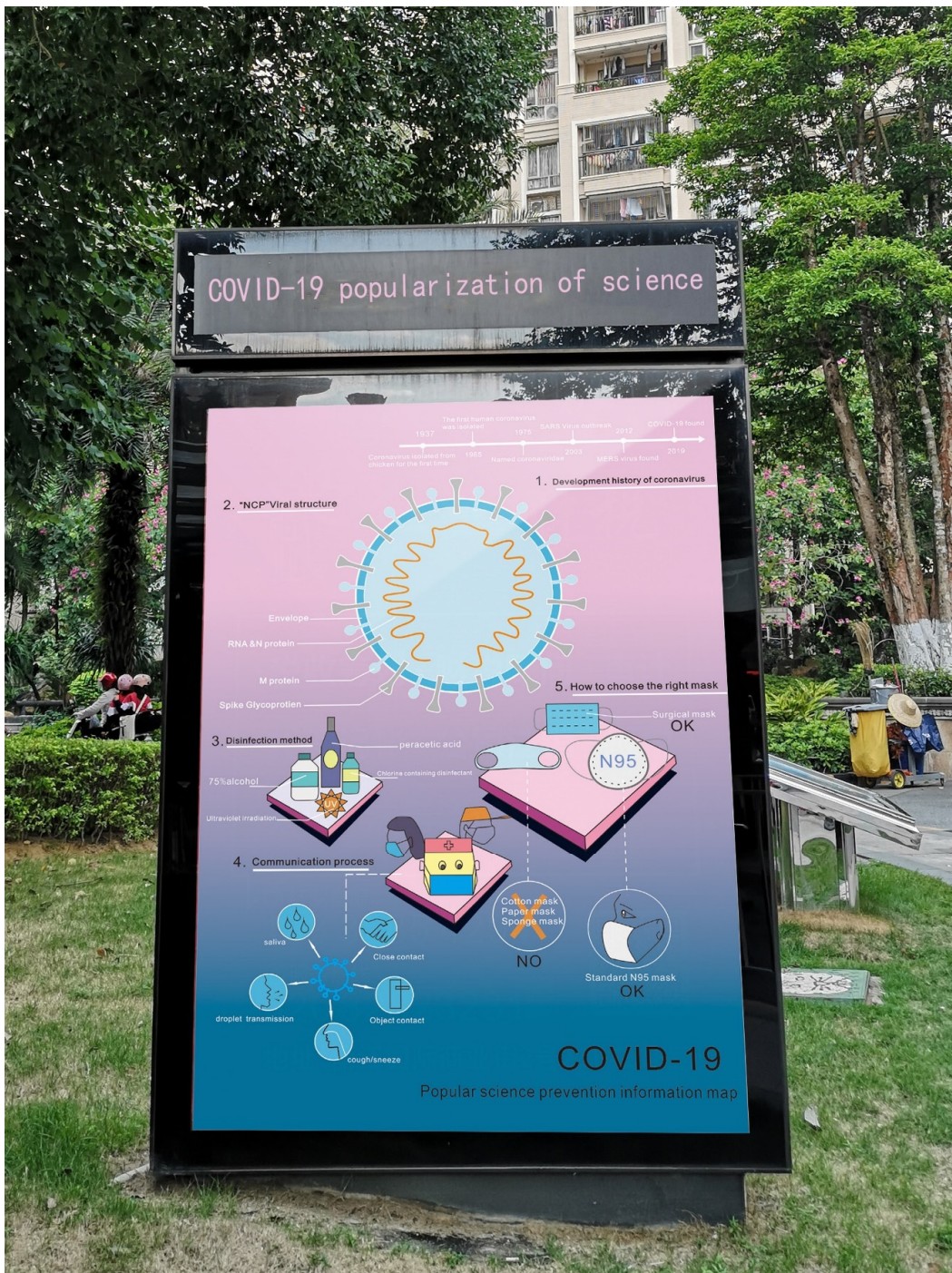

**Figure 15.** Community COVID-19 science popularization prevention infographic.

## 5. Conclusions

The effective participation of the public in the scientific knowledge popularization of public health events is an important element of national governance ability during public health crises. To improve the public's interest and attention to popular science [55], it is necessary to strengthen the dissemination of information about major public events and constantly improve the popular science information visualization design method. The perceptual narrative design method has become one of the best ways to connect information transmission with the audience and has been an important part of the process to realize science popularization information perception to cognition [56,57]. To further promote the

research on perceptual narrative in the field of popular science information visualization, the following issues should be noted:

First, more attention should be paid to public health events and the top-level design within the framework of the emergency response system needs to be strengthened. In major public health emergencies, a multiplatform joint early warning mechanism should be established to collect public related response problems and hot spots [58] and continuously improve the construction of emergency science popularization [59].

Second, narrative design puts forward enlightening empirical research in the field of product design, but there are relatively few theoretical models and practices that perceptual narrative research can refer to in emergency science popularization design [60,61], which means that it is necessary to conduct simple and constrained valuable research in perceptual narrative field, accurately handle the optimization relationship of emergency science popularization information elements and create innovative design models with perceptual narrative theory [62].

Third, for this study, there are still some deficiencies in terms of experiments. For example, the selected experimental popularization elements are not enough and the group sought in the perceptual research part is too small.

Research directions will be explored in the future from the following aspects: (1) Improve the method of mining public needs. The method used in this study should be combined with a KANO model to more accurately explore the public's demand for popular science information [63–65]. (2) Increase the number of popular science information elements and sufficient sample tests in the study to improve the accuracy of the design, which can further improve the match between popular science information design and public psychological needs. (3) Cooperation channels need to be broadened. Designers can work with public health experts, communication scholars and public opinion analysts to cross integrate statistics, iconography, computer vision, aesthetics, communication and other disciplines to study popular science information design methods and expand the scope of research application [66–68].

**Author Contributions:** H.L. designed the pilot and case study projects, conducted a user study, developed a framework, and drafted the manuscript. K.W. guided the study and assisted with editing. All authors have read and agreed to the published version of the manuscript.

**Funding:** Guangzhou Huashang College Intramural Mentorship Research Project (2021HSDS21).

**Institutional Review Board Statement:** Not applicable.

**Informed Consent Statement:** Not applicable.

**Data Availability Statement:** Not applicable.

**Acknowledgments:** The author would like to thank those who participated in and cooperated with this study for their help in the smooth development of this study.

**Conflicts of Interest:** The authors declare no conflict of interest.

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
