# Peer review of "Research on Design of Emergency Science Popularization Information Visualization for Public Health Events-Taking “COVID-19”as an Example"

_sustainability, doi:10.3390/su14074022_

Round 1

Reviewer 1 Report

Thank you for giving me the opportunity to review the paper. It's an interesting topic and I read it well. However, it is expected to be a more complete paper after a few supplements as follows.

  1. It would be better to mention the framework and order of this study in the introduction.

  1. I can see grammar errors, typos, spacing errors, etc. Please review the paper as a whole and revise it. For example, at the beginning of the first sentence of the introduction, and the interval between “The On January~” and “easy to” and “get around” in line 7.

  1. It will be easier for readers to read if you match the titles 1 to 5 specified in Figure 5 and the sub-headings presented from the bottom of the picture.

  1. Figure 8 and 9 overlap with the text, so it is impossible to check the contents. And even in the picture, there are cases where the font size is not unified or the line is not neatly processed, so correction is required. There is also a problem in showing the title of Table 3.

Reviewer 2 Report

    In the paper, the authors explores the optimization method of emergency popular science information design elements in public health events.

  But the research was not presented clearly. In my opinion, the paper mainly introduce the popular science information design process with the case of COVOD-19. So, what is the innovation? Covid-19 is a good case, so it is necessary to discuss according to the case. So, I suggest the authors to reconstructed the paper, and derive the academic topics.

   Moreover, the layout of the manuscript is poor, and I can’t read some important figures, e.g., Fig 8 and 9.  

Reviewer 3 Report

  1. This paper presents an emergency Science popularization Information visualization for public health events-taking “COVID-19”as an example. The idea is good. Anyway, there are some concerns in terms of contribution and presentation.
  2. The major contribution of this paper should be added in the last part of the introduction.
  3. Sections 1 and 2 need summarization. Also, section 2 does not need any introduction explanation, it should be shown the proposed approach.
  4. The details of the Fuzzy such as membership function, rules, fuzzifier, and defuzzifier should be explained exactly.
  5. Since Figures 1, 2, 3, 11, and 12 are from other research, they should be cited in the caption. Also, the texts inside these figures are not clear to read, is there any solution for it?
  6. Figures 12 and 13 are the same. One of them should be removed.
  7. In page 7, the authors wrote “When acquiring information, designers should use statistics or machine learning methods”. Could you explain why ML is required? What are your suggestions in terms of method, dataset?
  8. There are some methods in ML that are suitable for feature extraction. The authors need to explain why their approach is better than them. Also, a comparison with state-of-the-art research is required.
  9. Page 11 is not readable due to overlapping figures and texts.
  10. All assumptions and restrictions of this work can be explained.
  11. The future work directions should be explained in the last part of the paper.
  12. Some related works in 2022 should be added with their pros and cons.

Round 2

Reviewer 2 Report

  The authors have tried to revise the paper, and the opions and thoughs can be proposed clearly in the current version.

   I suggest to go through the paper, and correct somem errors in grammars, such as  The On January 30, 2020, Beijing time.

Reviewer 3 Report

1. Most of the comments have been addressed. There are some comments that improve the paper before publication.
2. Before starting the contribution in the introduction, write "The main contribution of this paper is". This helps readers to correctly follow your paper's idea.
3. All references related to figures should be added in the figures' captions.
4. All figures need a caption below them. Also, instead of "As shown in the figure below", refer to figure by figure number.
5. Figure 7 is not clear. The background color of all boxes should be removed.
6. Figure 9 is not readable completely. 
7. The Fuzzy system should have at least one basic or advanced reference.
8. The abstract and conclusion sections should be more summarized. Especially most parts of the conclusion should be moved to the discussion section.
9. Adding some related works in 2022 can support the novelty of this research.
